# Microbial Diversity and Volatile Flavor Compounds in Tibetan Flavor *Daqu*

**DOI:** 10.3390/foods12020324

**Published:** 2023-01-09

**Authors:** Yaping Li, Haijun Qiao, Rui Zhang, Weibing Zhang, Pengcheng Wen

**Affiliations:** 1College of Food Science and Engineering, Gansu Agricultural University, Lanzhou 730070, China; 2College of Science, Gansu Agricultural University, Lanzhou 730070, China; 3Functional Dairy Product Engineering Lab of Gansu Province, Yingmen Village, Anning, Lanzhou 730070, China

**Keywords:** TF *Daqu*, physicochemical indices, microbial diversity, volatile flavor compounds

## Abstract

In this study, the microbial diversity in Tibetan flavor *Daqu* was analyzed based on single molecule real-time sequencing (SMRT). The volatile flavor compounds in *Daqu* were detected using the headspace solid-phase microextraction-gas chromatography-mass spectrometry (HS-SPME-GC-MS). In addition, the correlation between various microbes and volatile flavor compounds was explored. Our studies indicated that the dominant bacterial genera in Tibetan flavor *Daqu* were *Oceanobacillus*, *Kroppenstedtia*, *Virgibacillus*, *Enterococcus*, *Pediococcus*, *Streptomyces*, *Saccharopolyspora*, *Leuconostoc*, *uncultured_bacterium_f_Lachnospiraceae* and *Lactobacillus*. The dominant fungal genera were *Wickerhamomyces*, *Monascus*, *Aspergillus* and *Rhizomucor*. 101 volatile compounds were detected in the *Daqu* samples, including alcohols, acids, esters, aldehydes, hydrocarbons, ketones, ethers, aromatics and pyrazines, and 10 key flavor compounds were identified using the relative odor activity value (ROAV). The results of our correlation analysis showed that *Enterococcus* was mainly associated with the synthesis of aldehydes such as trans-2-octenal, and *uncultured_bacterium_f_lachnospiraceae* was associated with the synthesis of most aldehydes. This paper has systematically investigated the physicochemical indices, microbial community structure and flavor compounds of Tibetan flavor *Daqu*, which is helpful in gaining a deeper understanding of the characteristics of Tibetan flavor *Daqu*.

## 1. Introduction

Liquor is a distilled spirit unique to China. There are many types of liquor, which include three main categories: Maotaiflavor liquor, Luzhou flavor liquor and Light flavor liquor [1]. In addition, there are some less well-known liquors that are also of excellent quality, such as Tibetan flavor liquor (TF liquor). TF liquor is a distilled liquor produced via the saccharification and fermentation of *Daqu* under certain conditions, and it is favored by Tibetans. TF liquor is pure with a natural and coordinated compound aroma dominated by ethyl caproate and supplemented by ethyl acetate. The taste of TF liquor is sweet and refreshing, and it is mellow and has a long remaining taste [2].

During the fermentation process of liquor, Jiuqu, also known as *Daqu*, plays a decisive role [1]. *Daqu* is the saccharifying and fermenting agent of liquor, which directly affects the flavor, quality and taste of liquor. *Daqu* is divided into many types according to different flavors, such as Maotai flavor *Daqu*, Luzhou flavor *Daqu* and Light flavor *Daqu* [1,2]. Tibetan flavor liquor is fermented with Tibetan flavor daqu (TF *Daqu*). TF *Daqu* is a medium-temperature *Daqu*; the process diagram of its production is shown in Figure 1. TF *Daqu* is made of high-quality barley, wheat, highland barley, peas and maize with crushed Tibetan medicinal materials. It is made via the traditional TF *Daqu* adobe house cultivation process and the top culture temperature is 35–48 °C. Its stubble is clear, hard and tidy with an intense fragrance of *Daqu* and Tibetan medicinal materials, and the storage period is over 4 months [2].

Because *Daqu* is naturally inoculated and produced via an open fermentation process, the microorganisms in *Daqu* are very complex [1,2]. To date, there have been many research studies on the functions and types of microorganisms in *Daqu*. In general, the microorganisms in *Daqu* can be mainly divided into four categories: bacteria, molds, yeasts and actinomycetes [3]. *Bacillus*, one of the most representative bacteria, can secrete a variety of degrading enzymes, such as protease, amylase and glycosylase, which will decompose proteins, starch and other macromolecular substances during the fermentation of liquor to produce a variety of flavor substances [3]. Lactic acid bacteria can not only synthesize extracellular polysaccharides and oligosaccharides, but also produce the aroma substances of liquor via the Maillard reaction and produce a large amount of lactic acid, which can enhance the sweetness and mellowness of the wine at the same time [4]. *Aspergillus* in *Daqu* can produce enzymes related to starch saccharification and protein hydrolysis, thus affecting the flavor of liquor, while playing a vital role in the saccharification and esterification abilities of *Daqu* during its fermentation [5]. *Rhizopus* can produce organic acids, including succinic acid, lactic acid and fumaric acid, which are crucial in the formation of the characteristic taste of liquor [6]. Yeast in *Daqu* is the primary microbe in the brewing process. During the manufacturing of liquor, *Saccharomyces cerevisiae* transforms glucose into alcohol and produces esters, higher alcohols and organic acids [6]. In addition, *Daqu* is also known as the aroma-producing agent of liquor. During the *Daqu* manufacturing process, the metabolites of the microorganisms and decomposition products of the raw materials directly or indirectly contribute to the flavor substances in liquor, giving it a variety of unique flavors [7]. At present, many studies have introduced common *Daqu* in detail, including Maotai flavor *Daqu*, Luzhou flavor *Daqu* and Light flavor *Daqu* [6,7,8], but there are few reports on TF *Daqu*. Therefore, exploring the microbial diversity and volatile flavor compounds of TF *Daqu* can help to enhance the quality of TF *Daqu*, which is of far-reaching significance in the brewing of liquor.

In this study, SMRT (single molecule real-time) and HS-SPME-GC-MS (headspace solid-phase microextraction-gas chromatography-mass spectrometry) were used to detect the microbial diversity and volatile flavor compounds in TF *Daqu*, respectively. The correlation between the microbial diversity and volatile flavor compounds was also analyzed. The results of the present study supply some theoretical foundation for the quality improvement of TF *Daqu* and provides theoretical support for further optimization of the microorganisms in liquor and product quality control.

## 2. Materials and Methods

### 2.1. Sample Collection and Pretreatment

The TF *Daqu* samples were collected from the Tibetan flavor liquor distillery in Tianzhu County, Gansu Province. TF *Daqu* blocks were randomly selected from five different batches. Samples from the same batch were crushed and mixed to form a mixed sample. Thus, a total of 5 mixed samples were formed and numbered ZQ1, ZQ2, ZQ3, ZQ4 and ZQ5. The samples were preserved at 4 °C to detect the flavor compounds, and stored at −80 °C for analysis of the microbial diversity.

### 2.2. DNA Extraction, PCR Amplification and High-Throughput Sequencing

A Power Soil DNA isolation kit was used to extract DNA from *Daqu*. The extracted DNA was then determined for its purity, concentration and integrity. To analyze the bacteria in the samples, the full-length 16S rRNA gene was amplified with 27F(5′-AGRGTTTGATYNTGGCTCAG-3′)/1492R(5′-TASGGHTACCTTGTTASGACTT-3′)primers. The amplification procedure was as follows: pre-denaturation at 95 °C for 2 min; denaturation at 98 °C for 10 s, annealing at 55 °C for 30 s, extension at 72 °C for 1.5 min, a total of 25 cycles; and extension at 72 °C for 2 min. ITS1F(5′-CTTGGTCATTTAGAGGAAGTAA-3′)/ITS4R(5′-TCCTCCGCTTATTGATATGC-3′)primers amplified the full-length ITS sequences of *Daqu* for fungi. The amplification procedure was as follows: pre-denaturation at 95 °C for 5 min, 8 cycles (denaturation at 95 °C for 30 s, annealing at 55 °C for 30 s, extension at 72 °C for 45 s), 24 cycles (denaturation at 95 °C for 30 s, annealing at 60 °C for 30 s, extension at 72 °C for 45 s), and extension at 72 °C for 5 min. The products were purified, quantified and homogenized, and the resulting library was established and quality checked. Finally, the qualified libraries were tested on the PacBio sequencing platform.

### 2.3. Analysis of the Microbial Diversity

The original sequences were derived and the effective sequences were obtained after identifying, filtering and removing the chimeras. The effective sequences were clustered according to their similarity and the OTU (operational taxonomic units) were divided. According to the results of our OTU analysis, the samples were analyzed taxonomically at various taxonomic levels. Alpha diversity analysis, which includes the Simpson, Shannon and Chao1 indices of the samples, was used to study the species diversity and richness. The microbial community structure was analyzed using the relative abundance of species at the phylum and genus level.

### 2.4. Analysis of the Volatile Flavor Compounds in Daqu Using HS-SPME-GC-MS

SPME fiber (50/30 μmDVB/CAR/PDMS; Supelco, Bellefonte, PA, USA) was used to sample volatile flavor compounds. A total of 3.00 g of the crushed *Daqu* sample was placed in a 15 mL headspace bottle and subsequently sealed. The bottle was then placed in a thermostat water bath heated at 60 °C for 15 min and the SPME fiber was inserted at 60 °C for 50 min to carry out the extraction. The extraction head was inserted into the GC-MS inlet and desorbed at 250 °C for 3 min for GC-MS analysis.

The volatile flavor compounds were detected using GC-MS (Trace GC 1310-ISQ mass spectrometer; Thermo Scientific, Austin, TX, USA). The GC-MS conditions used were in accordance with the method reported in a previous study with some slight modification [8]. A DB-5MS flexible quartz capillary column (30 m × 0.25 mm × 0.25 μm) was applied. The heating procedure was carried out as follows: The temperature was initially set at 40 °C, then increased to 150 °C at a rate of 5 °C/min for 3 min, and then heated to 250 °C at a rate of 5 °C/min for 2 min. High-purity helium at a flow rate of 1.0 mL/min was used as the carrier gas. The MS conditions were as follows: An EI source served as the ion source and its temperature was 230 °C, the temperature of the connecting port was 280 °C and the electron energy was 70 eV. The quality scanning range was 40–450 amu.

The obtained spectra were searched and analyzed using the Wiley spectral library provided with the instrument and the peaks with similarity of <80% and siloxane-type impurities were screened and removed, and the volatile compounds in the TF *Daqu* samples identified. The peak-area normalization method was used to determine the relative content of the compounds. The different flavor compounds were analyzed and compared using software such as Excel.

### 2.5. Identification of the Key Flavor Compounds in Daqu Using ROAV (Relative Odor Activity Value)

Referring to the method described by Cai et al. [9], ROAV was applied to determine the key flavor compounds in the Tibetan *Daqu* samples. The ROAV_max_ was defined as 100 for the volatile compound with the largest contribution to the aroma of the samples and the ROAVs of the other volatile compounds (A) were determined using the following formula:(1)$$\mathrm{ROAV}=\frac{C\%A}{C\%\max}\times \frac{\mathrm{Tmax}}{\mathrm{TA}}\times 100$$
where C%A and C%max denote the relative content of each aromatic component and the aromatic component with the largest contribution to the aroma, respectively. C%A and C%max were calculated by GC-MS. TA and Tmax denote the odor threshold of each aromatic component and the aromatic component with the largest contribution to the aroma, respectively. TA and Tmax were obtained by querying the odor threshold table.

### 2.6. Correlation Analysis of the Microbial Community Diversity and Flavor Compounds in TF Daqu

SPSS software was used to calculate correlation coefficients and *p*-values. The Spearman correlation coefficients and *p*-values were used to study the correlation between the dominant flora and key flavor compounds in *Daqu*. A significant correlation was defined as a correlation coefficient >0.6 and *p* < 0.05.

## 3. Results and Discussion

### 3.1. Sequencing Results and Alpha Diversity

The sample sequences were processed using statistical methods to obtain high-quality sequences for subsequent analysis. As shown in Table 1, the number of effective bacterial sequences in the five TF *Daqu* samples ranged from 5150 to 6321 and the number of effective sequences of fungi ranged from 6069 to 7530. In addition, the proportion of bacterial effective sequences exceeded 80% and the proportion of fungal effective sequences exceeded 90% in all of the *Daqu* samples.

The alpha diversity reflects the species diversity and richness of single samples. Among them, the Chao1 index is applied as a measure of species richness and the Simpson and Shannon index reflects the species diversity [10]. In addition, the coverage was calculated; the larger the coverage value, the higher the probability that the species will be detected in the samples. The alpha diversity indexes are shown in Table 1.

Among the five TF *Daqu* samples studied, the Chao1 index of the bacteria ranged from 37.429 to 95.375, the Simpson index was between 0.739 and 0.943, and the Shannon index was between 2.726 and 5.090. The Chao1 index, Simpson index and Shannon index of ZQ1 and ZQ5 were relatively high, indicating that the bacterial community richness and diversity were higher than those of the other samples studied. For fungi, the Chao1 index ranged from 14.000 to 48.091, the Simpson index was between 0.060 and 0.214, and the Shannon index was between 0.277 and 0.821. The Chao1 index, Simpson and Shannon index of ZQ3 were relatively high, indicating that the diversity and richness of fungal communities were higher than those of the other samples studied. The coverage of the sequencing for the five *Daqu* samples were all over 99%, meaning that the amount of data for this sequencing was sufficient.

### 3.2. Microbial Community Structure of TF Daqu

The sequencing results show that there were 16 phyla, 158 genera and 171 species of bacteria in the five TF *Daqu* samples. In addition, there were 5 phyla, 35 genera and 39 species of fungi in the samples (Appendix A).

The relative abundance of the bacterial communities was analyzed at the phylum level (Figure 2A). Firmicutes was predominant in the five *Daqu* samples, accounting for >50% of the total, followed by Proteobacteria and Actinomycetes. This is consistent with the findings reported in previous studies. Xie et al. [11] studied the bacteria community and dynamic succession of sesame flavor liquor *Daqu* and their findings indicated that the most predominant flora was Firmicutes. In addition, Tian et al. [12] reported that Firmicutes was the most important phylum using a metagenomics-based study on the diversity of bacterial communities in Shilixiang liquor *Daqu*.

The relative abundance of the fungal communities was analyzed at the phylum level (Figure 2B). Ascomycota was the main phylum among the five TF *Daqu* samples, its relative abundance ranging from 97.07 to 99.35%. This is consistent with the findings of Jiang et al. [13] during their study on the microbial flora and dynamic succession during the manufacture of Northern Jiang flavored *Daqu* liquor, which reported that the main fungus was also Ascomycota.

The relative abundance of the bacterial communities was analyzed at the genus level (Figure 3A). It is considered that the dominant genus is when the relative abundance is greater than 1%. In sample ZQ1, the dominant bacteria were *Oceanobacillus*, *Kroppenstedtia*, *Virgibacillus*, *Enterococcus* and *uncultured_bacterium_f_Lachnospiraceae*. In the samples ZQ2, ZQ3 and ZQ4, the dominant bacteria were *Oceanobacillus*, *Kroppenstedtia*, *Virgibacillus*, *Enterococcus*, *Pediococcus*, *Streptomyces*, *Saccharopolyspora*, *Leuconostoc* and *Lactobacillus*. In sample ZQ5, the dominant bacteria were *Oceanobacillus*, *Kroppenstedtia*, *Enterococcus* and *uncultured_bacterium_f_Lachnospiraceae*. Among them, the relative abundance of *Oceanobacillus* in ZQ4 was 45.71%, while it was only 15.60% in ZQ5. The relative abundance of *Virgibacillus* in ZQ3 was 13.92%, while it was only 0.60% in ZQ5. It can be seen that although the TF *Daqu* samples contain a large number of bacterial communities, the distribution of the dominant bacteria in each sample was not uniform. Chen et al. [14] found that the dominant bacterial groups in the middle of special flavor liquor *Daqu* were *Oceanobacillus*, *Kroppenstedtia, Lactobacillus* and *Bacillus*. During analysis of the microbial diversity in various types of high-temperature *Daqu*, Wang et al. [4] found that the dominant bacteria were composed of *Bacillus*, *Brevibacterium*, *Kroppenstedtia*, *Lentibacillus*, *Staphylococcus*, *Saccharopolyspora*, *Streptomyces* and *Thermoactinomycetes*. Cai et al. [15] found that the core bacterial flora of low-temperature *Daqu* was dominated by *Lactobacillus*, together with *Saccharomyces*, *Bacillus* and *Streptomyces*.

The relative abundance of fungal communities was analyzed at the genus level (Figure 3B). The unclassified fungi were not discussed. In sample ZQ1, ZQ3 and ZQ4, the dominant fungi were *Wickerhamomyces, Monascus, Aspergillus* and *Rhizomucor*, and in sample ZQ5, the dominant fungi was *Wickerhamomyces*. Hui et al. [16] used SMRT to identify the microbial characteristics of koji and reported that the main fungal genus was *Wickerhamomyces*. *Aspergillus* and *Rhizoctonia* were found to be the dominant genera in Maotai flavor *Daqu* [17].

Since TF *Daqu* is a medium-temperature *Daqu*, its fermentation temperature is ~30–50 °C. This environment is suitable for most microorganisms to survive, so the microorganism communities are more abundant. The findings of this study suggest that the number of bacteria in *Daqu* was more abundant than that of fungi, which was consistent with the results of previous studies [18]. Although the microorganisms in TF *Daqu* have been reported in previous *Daqu* liquors, there are some differences between the microbial composition of each *Daqu* sample when compared with previous studies, which may be caused by the differences in the raw materials, techniques and environment used for the production of *Daqu* (Appendix A) [18].

### 3.3. Volatile Flavor Compounds in TF Daqu

The volatile flavor components in TF *Daqu* were determined using HS-SPME-GC-MS. There were 101 kinds of volatile compounds detected in the TF *Daqu* samples studied, including alcohols, aldehydes, esters, acids, hydrocarbons, ketones, ethers, aromatics and pyrazines (Table 2).

Table 2 shows that the flavor compounds in TF *Daqu* mainly consisted of esters, hydrocarbons and alcohols, and their relative contents account for 37.14, 30.17 and 7.64% of the total flavor compounds contents, respectively, while the relative contents of the other kinds of flavor compounds were <7%.

Esters determine the flavor type of liquor. In this study, esters were the compounds with the highest content and relatively more types than the other flavor compounds found in the TF *Daqu* samples. Among all of the flavor compounds, the relative content of methyl hexadecanoate was the highest (9.87%), followed by methyl linoleate (9.49%). Fan et al. [19] reported that the volatile compounds were mainly esters and alcohols in Fen flavor *Daqu*. Le et al. [20] also reached a similar conclusion during their analysis of the flavor substances in Fen *Daqu*. The hydrocarbon content in TF *Daqu* was second only to that of esters, and the relative content of tetradecane and dodecane was higher, accounting for 8.99 and 5.65%, respectively. Alcohols are the primary compound in liquor. In addition, the types and content of alcohols in TF *Daqu* were also high. Among them, the relative content of phenethyl alcohol was the highest, accounting for 4.31%.

Sun et al. [21] found that alcohols, esters and pyrazines were the main fragrance substances of sauce flavor *Daqu*. Meanwhile He et al. [22] detected 60 volatile substances in strong flavor *Daqu*, of which there were 42 esters, so esters were the main aroma components in strong flavor *Daqu*. The difference in the composition of the flavor substances in *Daqu* makes each *Daqu* liquor have a different flavor; the reason for the difference is that the microorganisms and enzymes in *Daqu* have different compositions and ratios [23].

### 3.4. Identification of the Key Flavor Compounds in TF Daqu

By analyzing the odor threshold of flavor compounds (Table 3), it was found that the relative content of trans-2-nonenal in all of the flavor compounds accounted for 0.37% and that the odor threshold was 0.09 µg/m^3^, which contributes the most to the integral flavor of TF *Daqu*. There were 10 compounds with an ROAV ≥ 1 in the TF *Daqu* samples studied. These were 1-octen-3-ol, phenethyl alcohol, phenylacetaldehyde, (E)-2-octenal, nonanal, (E)-2-nonenal, decanal, methyl laurate, isovaleric acid and eugenol; they are considered the main flavor compounds in TF *Daqu*. There were 12 compounds with 0.1 ≤ ROAV < 1, namely 1-octanol, hexanal, benzaldehyde, methyl nonanoate, 2-methylbutyric acid, octanoic acid, D-Limonene, dodecane, octadecane, 2-nonanone, 2,3,5-trimethylpyrazine and 2-pentylfuran; these volatile compounds play a modifying role in the overall flavor of TF *Daqu*, among which the ROAV of 2-pentylfuran was 0.97. Therefore, it can be considered to have an essential modifying effect on the integral flavor of TF *Daqu*.

According to the ROAVs of the volatile flavor compounds, the key flavor compounds in TF *Daqu* were mainly composed of aldehydes and alcohols. These flavor compounds are usually produced by microorganisms in *Daqu* during the fermentation process or produced by microorganisms degrading starch compounds in the raw materials during glycolysis [24]. 1-Octen-3-ol has a fruity, grass and dusty flavor, and phenethyl alcohol and phenylacetaldehyde have a rosy aroma [25]. Trans-2-nonenal has an oily and grassy flavor [26]. Nonanal has the fragrance of grass and orange [27]. Trans-2-octenal has an oily odor [28]. These flavor compounds contribute to the unique flavor of TF *Daqu*.

### 3.5. Correlation Analysis of the Microbial Community Diversity and Flavor Compounds in TF Daqu

The correlation between the dominant flora and key flavor compounds of *Daqu* was analyzed using the Spearman correlation coefficients and *p*-values. The correlation heat map is shown in Figure 4.

Among the dominant bacterial genera, a negative correlation was observed between *Oceanobacillus* and trans-2-nonenal (*p* < 0.05). A positive correlation was observed between *Virgibacillus* and eugenol, and a negative correlation was found between *Virgibacillus* and decanal (*p* < 0.05). *Enterococcus* had a significant positive correlation with trans-2-octenal (*p* < 0.01), a positive correlation with trans-2-nonenal and decanal, and a negative correlation with methyl laurate (*p* < 0.05). *Pediococcus* exhibited a significant negative correlation with trans-2-octenal (*p* < 0.01), a negative correlation with trans-2-nonenal and decanal, and a positive correlation with methyl laurate (*p* < 0.05). *Streptomyces* had a positive correlation with eugenol and a negative correlation with decanal (*p* < 0.05). A negative correlation was observed between *Saccharopolyspora* and decanal (*p* < 0.05). *Leuconostoc* exhibited a considerable negative correlation with trans-2-nonenal (*p* < 0.01) and a negative correlation with trans-2-octenal (*p* < 0.05). *Uncultured_bacterium_f_Lachnospiraceae* was positively correlated with trans-2-octenal, trans-2-nonenal and decanal (*p* < 0.05). A positive correlation was observed between *Lactobacillus* and methyl laurate and a negative correlation with trans-2-octenal *p* < 0.05). Among the dominant fungi, a positive correlation was observed between *Wickerhamomyces* and 1-octene-3-ol (*p* < 0.05). A positive correlation was observed between *Fusarium* and nonanal (*p* < 0.05).

For these ten key flavor compounds, *Enterococcus* and *uncultured_bacterium_f_Lachnospiraceae* contributed the most to the flavor compounds. Figure 4 shows *Enterococcus* was associated with the synthesis of aldehydes such as trans-2-octenal. *Uncultured_bacterium_f_Lachnospiraceae* was associated with the synthesis of most of the aldehydes detected.

*Enterococcus* has good biosafety and probiotic properties, is often used to accelerate the fermentation process in fermentation production, and can hydrolyze proteins and esters to endow foods with good flavor and metabolize bacteriocins [29]. Lachnospiraceae may be a potential probiotic. All members of the Lachnospiraceae family have fermentation properties, of which some have strong hydrolyzing activities, such as pectate lyase, pectin methylesterase, α-amylase, xylanase and α-L-arabinofuranosidase [30]. *Oceanobacillus* are present in Maotai flavor *Daqu*, Luzhou flavor *Daqu* and Fen flavor *Daqu*. During the fermentation of liquor, *Oceanobacillus* are capable of producing various enzymes such as protease, amylase, cellulase and esterase. In addition, *Oceanobacillus* promote the esterification and saccharification ability of *Daqu* to some extent [31]. *Kroppenstedtia* is the dominant group of high-temperature sesame-flavored *Daqu* and its ability to secrete cellulase is strong, which contributes to the liquefaction ability of *Daqu* and the alcohol production in this liquor [32]. *Virgibacillus* are widely found in the natural environment and can utilize most carbohydrates as carbon and energy sources. In addition, *Virgibacillus* is able to produce exoenzymes, such as amylase, protease, inulinase and gelatinase [33]. *Pediococcus* is often considered to be spoilage bacteria in wine and has the ability to synthesize extracellular polysaccharides, which give the wine a sticky and thick texture. In addition, *Pediococcus* can develop a variety of enzymes that enable the production of desirable fragrant substances in wine [34]. *Streptomyces* is an important type of microorganism that is widespread in all kinds of ecosystems. *Streptomyces* has the capacity to secrete alkaline phosphatase, esterase and phosphate hydrolase, which are likely to have a significant role in the composition of the flavor components or precursors in Maotai flavor liquor [4]. *Saccharopolyspora* is mainly found in the natural surroundings and can produce important bioactive substances [35]. *Leuconostoc* is present in many different environments and is critical for the preparation and fermentation of many dairy products, vegetables and grains. In addition, it generally forms the buttery taste of dairy products [36]. *Lactobacillus* is considered a probiotic, which is beneficial to human health. In addition, it is present in some fermented food products where it is helpful in preservation, aroma and nutrition [36]. *Monascus* is widely present in *Daqu*, fermented grains, brewing mash, etc. *Monascus* has been widely used for the production of *Monascus* pigments used to color traditional foods [37]. In addition, *Monascus* can produce a variety of beneficial metabolites as a fermenting bacterium for traditional Chinese foods. *Aspergillus* is widely distributed in the environment and not only can use monosaccharides, but also produces numerous enzymes to decompose proteins, polysaccharides and other organic macromolecules. *Aspergillus* is also commonly used in food fermentation, for example, koji fermentations used in the production of soy sauce and miso [38]. The role that these species play in TF *Daqu* needs further study.

## 4. Conclusions

The results of our study have shown that microorganisms are highly abundant in TF *Daqu* and the bacterial diversity was higher than the fungal diversity. A variety of volatile flavor compounds were detected in TF *Daqu*, which mainly consist of esters, hydrocarbons and alcohols. The results of our correlation analysis showed that *Enterococcus* and *uncultured_bacterium_f_Lachnospiraceae* contribute more to the flavor compounds. Through the characterization of the microbial communities and flavor compounds of TF *Daqu*, our understanding of TF *Daqu* has deepened, providing a direction for further research on the formation of flavor compounds in liquor and the enhancement of *Daqu* quality. This study has a guiding significance for producing liquor and enhancing the quality of the final product.

## Figures and Tables

**Figure 1 foods-12-00324-f001:**
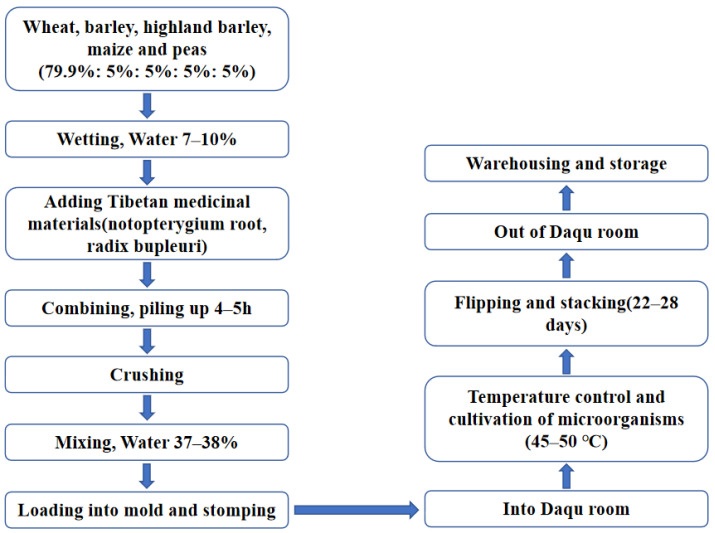
Process diagram for the production of Tibetan flavor *Daqu*.

**Figure 2 foods-12-00324-f002:**
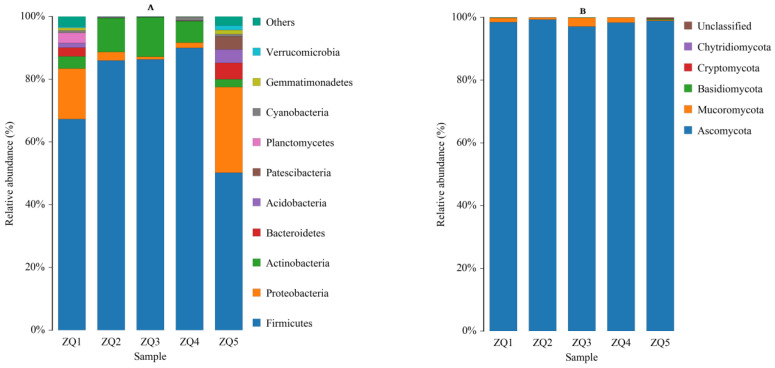
Microbial community structure in the samples at the phylum level: (**A**) Bacterial and (**B**) fungal community.

**Figure 3 foods-12-00324-f003:**
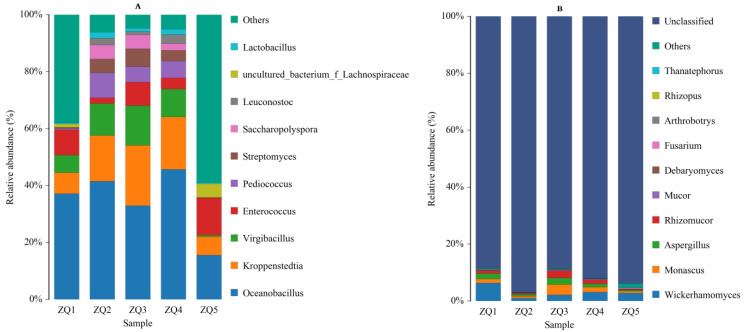
Microbial community structure in the samples at the genus level: (**A**) Bacterial and (**B**) Fungal community.

**Figure 4 foods-12-00324-f004:**
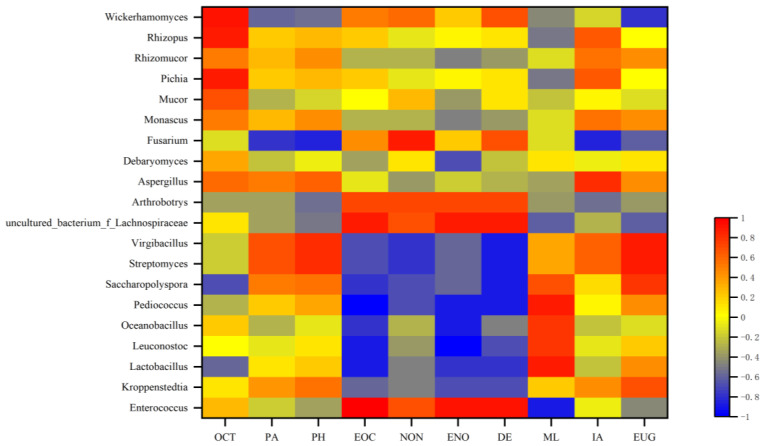
Correlation heatmap between the microbial community and 10 key flavor compounds: 1-Octen-3-ol (OCT), phenylethyl alcohol (PA), phenylacetaldehyde (PH), (E)-2-octenal (EOC), nonanal (NON), (E)-2-nonenal (ENO), decanal (DE), methyl laurate (ML), isovaleric acid (IA), andeugenol (EUG).

**Table 1 foods-12-00324-t001:** The sequencing results and the alpha diversity statistical analysis of TF Daqu samples.

	Bacteria					Fungi				
	Effective CCS	Chao1	Simpson	Shannon	Coverage	Effective CCS	Chao1	Simpson	Shannon	Coverage
ZQ1	6278	95.375	0.841	4.252	0.999	6069	18.000	0.213	0.766	0.999
ZQ2	5150	37.429	0.775	2.885	0.999	6600	14.000	0.060	0.277	0.999
ZQ3	6163	38.143	0.811	2.910	0.998	7410	17.500	0.214	0.821	0.999
ZQ4	6181	49.333	0.739	2.726	0.998	7114	20.333	0.153	0.604	0.999
ZQ5	6321	88.500	0.943	5.090	0.999	7530	48.091	0.148	0.711	0.999
Mean	6019	61.756	0.8218	3.5726	0.9986	6944.6	23.5848	0.1576	0.6358	0.999

**Table 2 foods-12-00324-t002:** Volatile flavor compounds identified in Tibetan flavor Daqu.

Number	Category	Compound Name	Retention Time (min)	CAS	ZQ1	ZQ2	ZQ3	ZQ4	ZQ5	Average
	Alcohols									
1		(2R,3R)-(-)-2,3-Butanediol	4.397	24347-58-8	-	-	0.74	1.44	0.77	0.59 ± 0.54
2		1-Octen-3-ol	9.507	3391-86-4	0.63	1.06	0.37	0.52	0.42	0.60 ± 0.24
3		3,5-Octadien-2-ol	11.335	69668-82-2	0.74	1.06	0.54	0.64	-	0.59 ± 0.34
4		1-Octanol	12.371	111-87-5	-	-	0.15	0.20	0.42	0.15 ± 0.15
5		1-Decanol	12.410	112-30-1	0.29	0.26	-	-	-	0.27 ± 0.01
6		Phenethyl Alcohol	13.730	60-12-8	1.82	3.44	5.63	7.36	3.30	4.31 ± 1.95
7		2-propyl-1-Heptanol	16.694	10042-59-8	0.18	-	-	0.33	0.43	0.18 ± 0.17
8		2-hexyl-1-Decanol	20.549	2425-77-6	0.22	0.14	-	-	-	0.18 ± 0.04
9		Palustrol	28.128	5986-49-2	0.18	0.56	0.15	0.24	0.25	0.27 ± 0.14
10		1-Hexadecanol	30.618	36653-82-4	-	-	0.11	0.57	-	0.13 ± 0.22
11		Ledol	31.098	577-27-5	-	1.65	-	-	-	0.33 ± 0.66
		Subtotal			4.06	8.17	7.69	11.30	5.59	7.63 ± 2.46
	Aldehydes									
12		Hexanal	4.797	66-25-1	0.96	1.74	0.97	1.39	1.54	1.32 ± 0.31
13		(E)-2-Heptenal	8.850	18829-55-5	-	0.16	0.11	0.14	0.22	0.12 ± 0.07
14		Benzaldehyde	9.033	100-52-7	0.69	0.99	0.75	1.04	1.01	0.89 ± 0.14
15		Phenylacetaldehyde	11.541	122-78-1	-	0.28	0.49	0.59	-	0.27 ± 0.24
16		(E)-2-Octenal	11.982	2548-87-0	0.39	0.54	0.38	0.46	0.65	0.48 ± 0.10
17		Nonanal	13.439	124-19-6	1.69	1.66	1.15	1.60	2.66	1.75 ± 0.49
18		(E)-2-Nonenal	15.173	18829-56-6	0.22	0.37	0.35	0.36	0.54	0.36 ± 0.10
19		Decanal	16.585	112-31-2	0.48	0.56	0.44	0.46	0.59	0.50 ± 0.05
20		2-Butyl-2-octenal	21.421	13019-16-4	-	2.59	-	-	-	0.51 ± 1.03
		Subtotal			4.43	8.89	4.64	6.04	7.21	6.24 ± 1.66
	Esters									
21		Methyl hexanoate	7.847	106-70-7	3.01	0.88	0.75	0.43	1.08	1.23 ± 0.91
22		Methyl heptanoate	10.863	106-73-0	0.57	0.18	0.25	0.06	-	0.21 ± 0.19
23		Methyl octanoate	14.012	111-11-5	2.95	1.97	2.29	1.70	2.39	2.26 ± 0.42
24		Octyl acetate	16.692	112-14-1	-	0.25	0.24	-	-	0.09 ± 0.12
25		Methyl nonanoate	17.083	1731-84-6	2.81	0.76	1.30	0.39	0.78	1.20 ± 0.85
26		Decanoic acid, methyl ester	20.030	110-42-9	1.13	0.81	1.30	0.68	0.82	0.94 ± 0.22
27		2(3H)-Furanone, dihydro-5-penty	21.087	104-61-0	-	-	0.26	0.27	0.40	0.18 ± 0.15
28		2-(2-Butoxyethoxy)ethyl acetate	21.089	124-17-4	0.77	0.77	-	-	-	0.30 ± 0.37
29		Ethyl (E)-4-decenoate	21.585	76649-16-6	-	0.39	0.29	0.42	-	0.22 ± 0.18
30		Heneicosanoic acid, methyl ester	21.989	6064-90-0	-	-	0.39	-	-	0.07 ± 0.15
31		Methyl laurate	26.297	111-82-0	0.49	-	0.65	-	-	0.22 ± 0.28
32		Methyl 10-Methylundecanoate	26.299	5129-56-6	-	-	-	0.29	0.47	0.38 ± 0.09
33		Methyl tetradecanoate	32.507	124-10-7	1.97	-	1.38	0.54	0.84	0.94 ± 0.67
34		Methyl pentadecanoate	34.172	7132-64-1	1.00	0.73	0.57	0.17	0.28	0.55 ± 0.30
35		Methyl oleate	34.371	112-62-9	-	-	0.31	-	-	0.06 ± 0.12
36		Diisobutyl phthalate	35.935	84-69-5	1.24	1.00	-	-	-	0.44 ± 0.55
37		Methyl hexadec-9-enoate	36.844	10030-74-7	0.82	-	0.53	0.20	-	0.31 ± 0.32
38		Methyl (Z)-hexadec-9-enoate	36.969	1120-25-8	0.54	0.35	1.14	0.46	-	0.49 ± 0.37
39		Methyl hexadecanoate	37.502	112-39-0	14.99	9.99	8.21	9.79	6.35	9.86 ± 2.87
40		Dibutyl phthalate	38.135	84-74-2	2.34	1.99	-	-	-	0.86 ± 1.06
41		Ethyl hexadecanoate	38.973	628-97-7	0.10	0.27	0.28	0.44	0.20	0.25 ± 0.11
42		Methyl linoleate	41.073	112-63-0	8.04	7.33	17.46	10.29	4.34	9.49 ± 4.41
43		Methyl trans-9-Octadecenoate	41.206	1937-62-8	7.21	4.90	9.62	5.46	3.19	6.07 ± 2.18
44		Methyl stearate	41.725	112-61-8	0.54	0.23	0.43	0.16	-	0.27 ± 0.19
45		Ethyl linoleate	42.386	544-35-4	-	0.17	0.23	0.30	-	0.14 ± 0.12
		Subtotal			50.52	32.97	47.88	32.05	21.14	37.14 ± 10.89
	Acids									
46		Isovaleric acid	6.219	503-74-2	0.00	0.32	0.19	0.45	0.00	0.19 ± 0.17
47		2-Methylbutyric acid	6.447	116-53-0	0.00	0.49	0.05	0.27	0.42	0.24 ± 0.19
48		Octanoic acid	15.756	124-7-2	0.00	0.11	0.00	0.07	0.00	0.03 ± 0.04
		Subtotal			0.00	0.92	0.24	0.79	0.42	0.47 ± 0.34
	Hydrocarbons									
49		1,3,5,7-Cyclooctatetraene	6.988	629-20-9	0.63	0.64	0.38	0.58	1.34	0.71 ± 0.32
50		D-Limonene	11.112	5989-27-5	0.69	0.73	0.56	0.67	1.86	0.90 ± 0.48
51		Undecane	13.320	1120-21-4	0.55	0.58	0.33	0.61	1.05	0.62 ± 0.23
52		9-methylheptadecane	15.504	26741-18-4	-	-	-	0.55	-	0.11 ± 0.22
53		3,8-Dimethyldecane	15.515	17312-55-9	-	0.32	-	0.23	-	0.11 ± 0.13
54		Dodecane	16.441	112-40-3	5.31	5.50	3.95	5.87	7.63	5.65 ± 1.18
55		2,6,10-Trimethyldodecane	16.798	3891-98-3	-	0.14	0.10	0.13	0.21	0.14 ± 0.04
56		1,7-Dioxaspiro[5.5]undec-2-ene	18.282	78013-58-8	-	0.37	0.24	0.34	-	0.31 ± 0.05
57		1-Tridecene	19.137	2437-56-1	-	-	-	0.56	-	0.11 ± 0.22
58		Tetradecane	19.429	629-59-4	8.62	8.96	6.63	8.61	12.11	8.98 ± 1.76
59		7-Methylheptadecane	20.351	20959-33-5	0.24	-	0.22	0.34	0.40	0.30 ± 0.07
60		Heneicosane	20.375	629-94-7	0.85	2.80	1.06	0.92	2.67	1.66 ± 0.88
61		3,5-Dimethyldodecane	20.452	107770-99-0	0.15	-	0.17	-	0.24	0.18 ± 0.03
62		(1-Propylnonyl)cyclohexane	20.694	13151-84-3	-	-	0.60	0.71	-	0.26 ± 0.32
63		2-Cyclohexyldodecane	20.697	13151-82-1	0.55	-	-	-	0.80	0.67 ± 0.12
64		2,6,11,15-Tetramethylhexadecane	21.241	504-44-9	0.27	0.35	0.23	0.34	0.45	0.32 ± 0.07
65		2-Methyltetracosane	21.391	1560-78-7	2.28	-	1.48	1.95	-	1.14 ± 0.96
66		α-Copaene	21.659	3856-25-5	-	0.12	0.11	0.13	0.20	0.14 ± 0.03
67		3-Methylidenetridecane	21.778	19780-34-8	0.29	-	0.29	0.40	0.52	0.30 ± 0.17
68		Longifolene	22.728	475-20-7	0.18	0.27	0.19	0.22	0.40	0.25 ± 0.08
69		Caryophyllene	23.001	87-44-5	1.23	3.55	0.56	0.48	0.90	1.34 ± 1.13
70		Decylcyclopentane	23.784	1795-21-7	0.41	0.40	0.39	0.53	0.67	0.48 ± 0.10
71		2,6,10-Trimethyltridecane	24.072	3891-99-4	0.16	0.16	0.15	0.16	-	0.15 ± 0.00
72		Cyclooctacosane	25.232	297-24-5	-	-	-	0.48	-	0.09 ± 0.19
73		Pentadecane	25.525	629-62-9	-	-	0.29	0.19	0.43	0.18 ± 0.16
74		Heptadecane	25.557	629-78-7	0.82	0.45	-	-	-	0.25 ± 0.33
75		8-Hexylpentadecane	26.581	13475-75-7	0.33	0.69	0.46	0.67	1.09	0.64 ± 0.25
76		Octadecane	26.734	593-45-3	0.13	-	-	0.19	-	0.16 ± 0.03
77		n-Nonylcyclohexane	27.442	2883-2-5	0.09	0.16	0.13	0.17	0.24	0.15 ± 0.04
78		Eicosane	27.750	112-95-8	0.26	0.94	0.56	0.67	2.53	0.99 ± 0.79
79		Phytane	34.686	638-36-8	0.21	0.26	0.15	0.09	0.51	0.24 ± 0.14
80		Dotriacontane	39.274	544-85-4	-	-	-	3.29	0.50	0.75 ± 1.28
81		Hexatriacontane	46.994	630-6-8	-	1.30	3.70	0.69	3.22	1.78 ± 1.43
		Subtotal			24.25	28.69	22.93	30.77	39.97	30.17 ± 6.04
	Ketones									
82		2-Nonanone	12.994	821-55-6	0.36	0.39	0.29	0.39	0.87	0.46 ± 0.20
83		6-Dodecanone	18.498	6064-27-3	0.25	-	0.17	0.22	-	0.12 ± 0.10
84		6-Undecanone	18.518	927-49-1	-	0.32	-	-	-	0.06 ± 0.12
85		trans-3-Nonen-2-one	20.530	18402-83-0	-	0.27	0.35	0.44	-	0.21 ± 0.18
86		(Z)-Oxacyclopentadec-6-en-2-one	30.271	63958-52-1	0.92	1.04	1.58	2.27	1.27	1.41 ± 0.48
87		6,10,14-Trimethyl-2-pentadecanon	35.525	502-69-2	0.17	0.25	0.17	0.17	0.25	0.20 ± 0.03
		Subtotal			1.70	2.27	2.56	3.49	2.39	2.48 ± 0.58
	Ethers									
88		1,2-Dimethoxybenzene	14.621	91-16-7	0.10	0.21	0.09	0.14	0.16	0.14 ± 0.04
89		5-Isopropyl-2-methylanisole	17.572	6379-73-3	-	0.10	0.09	0.12	0.19	0.10 ± 0.06
		Subtotal			0.10	0.31	0.18	0.26	0.35	0.24 ± 0.09
	Aromatics									
90		p-Cymene	10.907	99-87-6	-	-	0.24	0.27	-	0.10 ± 0.12
91		o-Cymene	10.954	527-84-4	0.31	0.25	-	-	0.70	0.42 ± 0.19
92		Eugenol	12.371	97-53-0	-	-	0.24	0.37	-	0.12 ± 0.15
93		Naphthalene	16.015	91-20-3	0.38	0.47	0.26	-	0.49	0.32 ± 0.17
94		4-Ethyl-1,2-dimethoxybenzene	19.904	5888-51-7	0.32	0.55	0.34	0.46	-	0.33 ± 0.18
95		Butylated Hydroxytoluene	25.654	128-37-0	2.61	5.64	3.07	4.22	6.33	4.37 ± 1.43
		Subtotal			3.62	6.91	4.15	5.32	7.52	5.67 ± 1.51
	Pyrazines									
96		2,5-Dimethyl pyrazine	7.503	123-32-0	-	-	0.13	0.21	-	0.06 ± 0.08
97		2,3,5-Trimethylpyrazine	10.201	14667-55-1	-	0.33	0.60	0.64	-	0.31 ± 0.27
98		Tetramethylpyrazine	12.798	1124-11-4	3.16	2.05	3.05	4.22	1.14	2.72 ± 1.04
99		2-Ethyl-3,5,6-trimethylpyrazine	15.041	17398-16-2	0.21	0.12	0.14	0.20	-	0.13 ± 0.07
		Subtotal			3.37	2.50	3.92	5.27	1.14	3.24 ± 1.38
	Others									
100		2-Pentylfuran	9.810	3777-69-3	1.19	0.22	0.45	0.56	1.35	0.75 ± 0.43
101		2-Acetylpyrrole	12.221	1072-83-9	-	-	0.21	0.31	-	0.10 ± 0.13
		Subtotal			1.19	0.22	0.66	0.87	1.35	

Note: “-” not perceived.

**Table 3 foods-12-00324-t003:** The ROAV of volatile flavor compounds in Tibetan flavor Daqu.

Number	Category	Compound Name	CAS	Odor Threshold (μg/m^3^)	ROAV
	Alcohols				
1		1-Octen-3-ol	3391-86-4	1	14.59
2		1-Octanol	111-87-5	22	0.17
3		Phenethyl Alcohol	60-12-8	12	8.74
	Aldehydes				
4		Hexanal	66-25-1	230	0.14
5		(E)-2-Heptenal	18829-55-5	2800	-
6		Benzaldehyde	100-52-7	85	0.26
7		Phenylacetaldehyde	122-78-1	1.7	3.89
8		(E)-2-Octenal	2548-87-0	2.7	4.36
9		Nonanal	124-19-6	3.1	13.75
10		(E)-2-Nonenal	18829-56-6	0.09	100.00
11		Decanal	112-31-2	2.6	4.73
	Esters				
12		Methyl heptanoate	106-73-0	290	0.02
13		Octyl acetate	112-14-1	140	0.02
14		Methyl nonanoate	1731-84-6	40	0.73
15		Dibutyl phthalate	84-74-2	260	0.08
16		Methyl laurate	111-82-0	1.5	3.70
	Acids				
17		Isovaleric acid	503-74-2	1.8	2.59
18		2-Methylbutyric acid	116-53-0	20	0.30
19		Octanoic acid	124-7-2	5.1	0.17
	Hydrocarbons				
20		D-Limonene	5989-27-5	45	0.49
21		Undecane	1120-21-4	5600	-
22		Dodecane	112-40-3	770	0.18
23		Tetradecane	629-59-4	5000	0.04
24		Caryophyllene	87-44-5	11000	-
25		Octadecane	593-45-3	20	0.19
	Ketones				
26		2-Nonanone	821-55-6	32	0.35
	Aromatics				
27		Naphthalene	91-20-3	450	0.02
28		p-Cymene	99-87-6	7200	-
29		Eugenol	97-53-0	0.61	4.86
	Pyrazines				
30		2,3,5-Trimethylpyrazine	14667-55-1	50	0.15
31		Tetramethylpyrazine	1124-11-4	2000	0.03
32		2,5-Dimethyl pyrazine	123-32-0	1820	-
	Others				
33		2-Pentylfuran	3777-69-3	19	0.97
34		2-Acetylpyrrole	1072-83-9	2000	-

Note: “-” not perceived.

## Data Availability

The data of microbial diversity has been submitted to the SRA of NCBI, and the bioproject number is PRJNA898104. Other data are available from the authors.

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
