# Peer review of "Microbial Diversity and Volatile Flavor Compounds in Tibetan Flavor Daqu"

_foods, 2023, doi:10.3390/foods12020324_

Round 1

Reviewer 1 Report

The article (reference number foods-2062287) entitled “Microbial diversity and volatile flavor compounds in Tibetan flavor Daqu” aims at investigating the profiles of microorganism and volatile compounds in the starter of a Chinese liquor. The characterization of microorganism is conducted based on varying phyla, genera, and species of bacteria and fungi. Large numbers of volatile compounds are identified, among which the key flavor components are pinpointed. The study also investigates the correlation between identified microorganism and key flavor components.

The research topic is quite interesting for both academy and food industry. However, the whole study is presented poorly in the most case, which makes the results questionable. Important information is missing from the description of methods. The results should be described more precisely. Authors failed to explain the reason causing the variation in microorganism and volatile compounds among five samples. In the section of discussion, the differences in study samples and methods among different studies should be highlighted when the results of the present study are compared with those from previous studies. There are also many mistakes when authors describe the results of previous studies. Moreover, authors should give a proper conclusion is needed instead of repeating the results again.

The specific comments are given as the following.

INTRODUCTION

The word “liquor” should not be capitalized since it is not a term.

Line 30-32 “There are many types of Liquor, which include three main categories: Maotai flavor Liquor, Luzhou flavor Liquor and Fen flavor Liquor”.  There is no such information in reference literature 1. Please check it.

Line 37-38 Please rephrase the sentence “its body is harmonious, soft and mellow with a long finish, and it is pleasant to drink, which leaves an aroma that is refreshing the next day”. Also, no such information is found in reference literature 2. Please check it.

Line 49 replace “>” with “over”

Line 52 delete “itself”

Line 78 provide the full names of “SMRT” and “HS-SPME-GC-MS”

MATERIALS AND METHODS

Line 86-92 The information of preparing sample preparation should be provided in detail. For example, what are the proportion of wheat, barley, highland barely, maize, and peas? what are the Tibetan medicinal materials? what is the condition of TF Daqu production?

The samples were collected from five batches. What is the difference among these batches? If all TF Daqu samples were produced at same condition, why did the samples show a large deviation in the profiles of microorganism and volatiles?

Line 94-100 Please provide gene sequences of the applied primers and also describe how PCR amplification was analyzed.

Line 104 The full name of OUT should be given.

Line 134 The full name of ROAV should be given.

Line 134-143 Please describe how the values of C%A, C%max, Tmax, and TA were measured.

Line 145 Please provide the name of software for calculating Spearman correlation coefficient.

RESULTS AND DISCUSSION

Line 151-155 Please clarify where the data was from.

Table 1 How many replicates were used for the analysis? Why there was only one value in the category of each sample? How was “coverage” measured? Please explain them.

Line 171-177 Sample comparison among samples as such is not appropriate. It needs a significance test, since some samples showed similar values.  

Line 181-183 Please clarify where the data was from.

Line 185-187 “Three dominant phyla were observed, including Firmicutes, Proteobacteria and Actinobacteria, and the total relative abundance of these three phyla ranged from 80.02 to 99.78% in the five Daqu samples studied.” Based on Figure 2A, I do not think Proteobacteria and Actinobacteria are dominant in all samples. Please rephrase it.

Line 189-191 “Xie et al. [11] studied the bacteria community and dynamic succession of sesame flavor Liquor Daqu and their findings indicated that the most predominant flora were Firmicutes, Proteobacteria and Actinomycetes.” In the study of Xie et al., Firmicutes is the only group pre-dominated in the samples. Please check it.

Line 205-207 “There were 10 dominant bacterial genera, Oceanobacillus, Kroppenstedtia, Virgibacillus, Enterococcus, Pediococcus, Streptomyces, Saccharopolyspora, Leuconosto, uncultured_bacterium_f_ Lachnospiraceae and Lactobacillus.” According to Figure 3A, they are not dominant in ZQ5 sample. Please rephrase it.

Line 211-213 “Chen et al. [14] found that the dominant bacterial groups in the middle of special flavor Liquor Daqu were Oceanobacillus, Kroppenstedtia and Bacillus.” It should be “Oceanobacillus, Kroppenstedtia, Lactobacillus, and Bacillus”. Please correct it.

Line 214-216 “Wang et al. [15] found that the dominant bacteria were composed of Bacillus, Brevibacterium, Kroppenstedtia, Lentibacillus, Staphylococcus, Saccharopolyspora, Streptomyces and Thermoactinomycetes.” I could not find this information in the study of Wang et al. Please check it.

Line 223-234 “The dominant fungal genera were Wickerhamomyces, Monascus, Aspergillus and Rhizomucor, and the relative abundance of Wickerhamomyces ranged from 0.97 to 6.35%.” Why were they the dominant fungal genera? Based on Figure 3b, most of fungal genera in the samples are unclassified?!

Line 225-226 Why did author cite a study regarding koji. This is not relevant.

Line 230-231 “The findings of this study suggest that the number of bacteria in Daqu was more abundant than fungi” Please clarify what ground the statement is based on.

Line 234-236 “which may be caused by the differences in the raw materials, techniques and environment used for the production of Daqu” Please clarify what “differences in the raw materials, techniques and environment for the production of Daqu” between your study and previous research.

Table 2  Regrading MS identification of volatile compounds, please add the information of retention indices of volatiles and base peak of mass spectra in the table. Moreover, please clarify what the data of ZQ1-5 mean. A significance test is also necessary if author compare the variation among samples.

Line 245 “their relative contents account for 37.14, 30.17 and 7.64%” Where are these data from? It they are the mean values calculated based on 5 samples, please add a column in Table 2 and express the values as mean ± standard deviation.  

Line 248 change “Liquor” to “liquor”

Line 251 “methyl hexadecanoate was the highest (9.87%), followed by methyl linoleate (9.49%)” Please clarify where the data from.

Line 255-256 “accounting for 8.99 and 5.65% ” Please clarify where the data from.

Line 266-289 Why only 34 compounds were calculated for key flavor compounds, since authors identified 101 volatile compounds from the samples? Please explain it.

Line 268 “trans-2-nonenal in all of the flavor compounds accounted for 0.37%” Please clarify where the data from.

Line 316 change “As a whole” as “For these ten key flavor compounds”

Line 317 “Fig. 3” should be Figure 4

Line 320-357 These discussion is not relevant to the correlation between microorganism and flavor compounds. Please delete this paragraph.

Reviewer 2 Report

The manuscript “Microbial diversity and volatile flavor compounds in Tibetan flavor Daqu” is important for the locality where the Tibetan flavor Liquor (TF Liquor) is produced. with an insight into the methodology and statistics, the work is of exceptional quality. The results obtained are interesting and excellently cross-referenced and explained. I suggest a minor revision, which is of a more technical nature:

1. line 15: “Study indicated” instead of “Our studies indicated”

2. lines 73-75: in the sentence “At present, many studies have introduced …” after “including Maotaiflavor Daqu, Luzhouflavor Daqu and Fenflavor Daqu” add reference.

3. line 128: Why did you chose exactly this value (80%)? Why not higher percentage?

4. lines 251-253: The references 20 and 21 are red in the text?

5. lines 320-355 are not results of this study. so this text should be moved to some more appropriate place, e.g. to introduction.

Author Response

Dear reviewers,

We sincerely thank you for taking your valuable time to revise the paper for us and giving us a golden opportunity to revise our manuscript. We appreciate the editor and reviewers very much for the positive comments and suggestions on our manuscript.

We sincerely thank you and the reviewer for your insightful and helpful comments, corrections, and suggestions on our manuscript, as well as the important guiding significance to our researches. We have studied comments carefully and examined the original manuscript and made many careful modifications according to the comments from the editor and the reviewers. We sincerely hope that the editor and reviewers can be satisfied with our responses. We believe that our work has benefited substantially from the invaluable input of the review team. The attachment is the detail of how we have addressed the review’s comments. The texts of our responses are marked in red (modifications based on questions from editors and reviewers) in this file. Attached please find the revised version, which we would like to submit for your kind consideration.

We would like to express our great appreciation to you and reviewers for comments on our paper. Looking forward to hearing from you.

Yours sincerely,

Weibing Zhang

Reviewer 3 Report

Chinese liquor is one of the world's oldest distilled alcoholic beverages, and it is typically obtained with the use of Daqufermentation starters. Daqu is a saccharifying and fermenting agent, having a significant impact on the flavour of the product. Although there is a wealth of artisanal experience in producing a range of different types of Daqu, the scientific knowledge base—including the microbiota, their enzymes and their metabolic activities—still needs to be developed. The experiments are well-designed, and the results support the conclusions. There are only a few points that need to be completed in the manuscript:

1.       Do the Authors optimize the used headspace solid-phase microextraction (HS-SPME) and gas chromatography-mass spectrometry (GC-MS) method?

2.       What statistical analysis was conducted to evaluate significant differences in volatile compounds? Such data should be added to the presented Tables.  What about the „n” value?

3.       What was the daqu processing temperature? What about daqu processing temperature effect on the fungal community and the quality of TF Daqu liquor?

4.       What about the measurement of pH and total titratable acidity (TTA)? The accumulation of acidic substances provides the ideal environment for the growth of microorganisms in daqu, affecting its microbial communities. Are there any data available?

5.       There are also some spelling errors in the text (for example, lines 65, 86, 94, etc.)

6.       Please add references in line 75.  

Author Response

Dear reviewers,

We sincerely thank you for taking your valuable time to revise the paper for us and giving us a golden opportunity to revise our manuscript. We appreciate the editor and reviewers very much for the positive comments and suggestions on our manuscript.

We sincerely thank you and the reviewer for your insightful and helpful comments, corrections, and suggestions on our manuscript, as well as the important guiding significance to our researches. We have studied comments carefully and examined the original manuscript and made many careful modifications according to the comments from the editor and the reviewers. We sincerely hope that the editor and reviewers can be satisfied with our responses. We believe that our work has benefited substantially from the invaluable input of the review team. The attachment is the detail of how we have addressed the review’s comments. The texts of our responses are marked in red (modifications based on questions from editors and reviewers) in this file. 

We would like to express our great appreciation to you and reviewers for comments on our paper. Looking forward to hearing from you.

Yours sincerely,

Weibing Zhang

Round 2

Reviewer 1 Report

All my questions have been answered. 

There is only one minor thing. Please remember to cite "Gemert, L.J.V. (2003) Compilations of Odour Threshold Values in Air, Water and Other Media. Zeist, The Netherlands:Oliemans Punter and Partners BV" (that was listed in your responce to reviewers) when you discribed "the odor threshold table" in the method section (line 189).